# CLASSICAL CIRCUITS CAN SIMULATE QUANTUM ASPECTS

M. Caruso

*Fundación I+D del Software Libre,* FIDESOL, *Granada (18100), Spain*[*]
*Universidad de Granada -* UGR, *Granada (18071), Spain.*[†] *and*
*Universidad Internacional de La Rioja -* UNIR, *Spain*[‡]

This study introduces a method for simulating quantum systems using electrical networks. Our approach leverages a generalized similarity transformation, which connects different Hamiltonians, enabling well-defined paths for quantum system simulation using classical circuits. By synthesizing interaction networks, we accurately simulate quantum systems of varying complexity, from $2-$state to $n-$state systems. Unlike quantum computers, classical approaches do not require stringent conditions, making them more accessible for practical implementation. Our reinterpretation of Born's rule in the context of electrical circuit simulations offers a perspective on quantum phenomena.

## I. INTRODUCTION

Recently, a procedure was established for connecting two quantum systems within a finite-dimensional Hilbert space through local transformations [1]. This correspondence provides a valuable tool for mapping quantum systems, enabling the study of one system through the lens of the other. This allows us to state that quantum systems described using finite-dimensional Hilbert spaces are, in principle, intersimulatable. By this, it is meant that it has been proven that there exists a transformation linking one to the other.

The dynamics of such quantum systems are governed by the Schrödinger equation, and a general recipe for constructing a classical electrical network capable of finding a solution of the Schrödinger equation from certain electrical signals has been described. The intention in this paper is to show how this can be done.

A productive and promising application for these ideas lies in the field of *quantum simulations.* Quantum simulators are controllable systems designed to emulate the static or dynamic properties of other quantum systems [3]. By establishing a gauge transformation, a connection between any two quantum systems residing in respective finite-dimensional Hilbert spaces can be established. In our work, the topic of quantum simulation was approached from an alternative perspective, starting with a more familiar scenario where the latter can be analytically solvable and/or simulatable. The first part of the approach is described in [2]. However, in this work, it will be shown that the time evolution, *a la* Schrödinger, of such quantum systems can be realized by a classical electrical network as an *analog simulator.* One approach to realize simulations of these equivalent quantum systems is through classical electrical networks. A classical system will be constructed in which certain dynamic quantities (voltages or currents) will evolve in time as does the wave function of a given quantum system, on a Hilbert space of finite dimension. This constitutes what is called an analog simulation of such systems.

There is a wide range of references to previous work in the field of classical simulata-

---

[*] mcaruso@fidesol.org
[†] mcaruso@ugr.es
[‡] mariano.caruso@unir.net

bility of quantum systems [4–6]. In all these cases, the intention is to simulate quantum gates in terms of optical devices. However, in this work, the aim is to make a classical simulation of the time evolution of a quantum system using electrical circuits.

A detailed and generalized description of the ideas proposed in [7, 8] for simulating quantum systems using specific circuits is provided. It is demonstrated that classical systems, which possess a simpler controllability compared to general quantum systems, can be utilized to accurately capture their temporal evolution. By leveraging this equivalence between quantum systems, the feasibility of implementing a comprehensive simulation protocol using classical circuits is established.

## II. QUANTUM SYSTEMS ON A FINITE DIMENSIONAL HILBERT SPACE

A brief introduction to the fundamental elements of quantum systems is provided. Consider a general quantum system $\mathbb{Q}$, which can be described in an $n$-dimensional Hilbert space. The deterministic temporal evolution of the quantum system is governed by a Hamiltonian operator $H$, which may be time-dependent. This operator acts on the vector state $|\psi(t)\rangle$ at time $t \in \mathcal{T} \subseteq \mathbb{R}$, as dictated by the Schrödinger equation $i\partial_t|\psi(t)\rangle = H|\psi(t)\rangle$, where $\partial_t$ denotes the partial derivative with respect to time and natural units, such as $\hbar = 1$, are employed. It is worth noting that a partial time derivative is used because $|\psi(t)\rangle$ may depend on other quantities. Alternatively, the Schrödinger equation can be expressed in terms of a particular basis $\boldsymbol{\beta} = \{|\beta_k\rangle\}_{k \in \mathcal{I}_n}$ as $\psi_k(t) := \langle\beta_k|\psi(t)\rangle$, where $k \in \mathcal{I}_n = \{1, \cdots, n\} \subset \mathbb{N}$. The bra-ket notation is utilized to represent the inner product [9], enabling the expression of the Schrödinger equation in terms of elements of $\mathbb{C}^n$, denoted as $\boldsymbol{\psi} = (\psi_1, \cdots, \psi_k, \cdots, \psi_n)^{\mathrm{t}}$. Note that all vectors presented in this work are of column type, the superscript $\mathrm{t}$ is used for matrix transposition, but for simplicity in notation, the notation is kept simple.

The complex vector$-$curve $\boldsymbol{\psi}$ satisfies another version of the Scrhödinger equation, given by

$$i\dot{\boldsymbol{\psi}}(t) = \boldsymbol{H}\boldsymbol{\psi}(t), \tag{1}$$

where $\dot{\boldsymbol{\psi}}$ denote the time derivative of $\boldsymbol{\psi}(t)$ and $\boldsymbol{H} \in \mathbb{C}^{n \times n}$ is a complex matrix that represents the hamiltonian operator $H$ in the basis $\boldsymbol{\beta}$ and whose matrix elements are $H_{kl} = \langle\beta_k|H|\beta_l\rangle$. The hamiltonian operator is referred to as the hamiltonian matrix.

## III. CLASSICAL SYSTEMS EQUIVALENTS TO QUANTUM SYSTEMS

In prior studies, Rosner [10] presented a compelling parallel between a specific quantum system and a classical system comprising electrical oscillators. This proposal was subsequently formalized and empirically demonstrated in [7, 8]. The purpose of this paper is to extend and generalize this formalization, encompassing quantum systems defined on any finite-dimensional Hilbert space.

Without loss of generality, a time-independent and self-adjoint Hamiltonian operator can be considered, represented by a constant Hermitian matrix $\boldsymbol{H}$.

Let us begin by expressing (1) using the *decomplexification* procedure, commonly referred to as *realification*, introduced by Arnold [11, 12]. This procedure involves mapping the

complex space $\mathbb{C}^n$ onto the real space $\mathbb{R}^{2n}$ using the linear operator $\mathfrak{D}\colon\mathbb{C}^n\longrightarrow\mathbb{R}^{2n}$. Note that $\mathfrak{D}$ is a linear operator and take a vector in $\mathbb{C}^n$ and return a vector in $\mathbb{R}^{2n}$ formed by the juxtaposition of the real and imaginary part of this vector, explicitly defined as $\mathfrak{D}(\boldsymbol{\psi}):=\big(\mathrm{Re}(\boldsymbol{\psi}),\mathrm{Im}(\boldsymbol{\psi})\big)$.

The equation (1) take the form of two separate equation for real and imaginary part of $\boldsymbol{\psi}$, denoted by $\boldsymbol{\varphi}_1 = \mathrm{Re}(\boldsymbol{\psi})$ and $\boldsymbol{\varphi}_2 = \mathrm{Im}(\boldsymbol{\psi})$

$$\dot{\boldsymbol{\varphi}}_1 = \boldsymbol{H}_2\,\boldsymbol{\varphi}_1 + \boldsymbol{H}_1\,\boldsymbol{\varphi}_2 \tag{2}$$

$$\dot{\boldsymbol{\varphi}}_2 = \boldsymbol{H}_2\,\boldsymbol{\varphi}_2 - \boldsymbol{H}_1\,\boldsymbol{\varphi}_1, \tag{3}$$

where $\boldsymbol{H}_1{=}\mathrm{Re}(\boldsymbol{H})$ and $\boldsymbol{H}_2{=}\mathrm{Im}(\boldsymbol{H})$, also the *dots* notations refers to time derivative. Note that the above equations can be obtained according to

$$\mathfrak{D}(\boldsymbol{H}\boldsymbol{\psi}){=}\big(\boldsymbol{H}_1\,\boldsymbol{\varphi}_1{-}\boldsymbol{H}_2\,\boldsymbol{\varphi}_2, \boldsymbol{H}_2\,\boldsymbol{\varphi}_1{+}\boldsymbol{H}_1\,\boldsymbol{\varphi}_2\big) \tag{4}$$

in particular $\mathfrak{D}(i\boldsymbol{\psi}){=}(-\boldsymbol{\varphi}_2\,,\boldsymbol{\varphi}_1)$.

The previous system of equations is decoupled by deriving both expressions.

$$\ddot{\boldsymbol{\varphi}}_1 = \boldsymbol{H}_2\,\dot{\boldsymbol{\varphi}}_1 + \boldsymbol{H}_1\,\dot{\boldsymbol{\varphi}}_2 \tag{5}$$

$$\ddot{\boldsymbol{\varphi}}_2 = \boldsymbol{H}_2\,\dot{\boldsymbol{\varphi}}_2 - \boldsymbol{H}_1\,\dot{\boldsymbol{\varphi}}_1, \tag{6}$$

clearing $\dot{\boldsymbol{\varphi}}_2$ from (3) and $\boldsymbol{\varphi}_2$ form (2) and replace them in expression (5) and the speculate proceeding solving $\dot{\boldsymbol{\varphi}}_1$ from (2) and $\boldsymbol{\varphi}_1$ form (3) and replace them in expression (6), to obtain

$$\ddot{\boldsymbol{\varphi}}_l(t) + \boldsymbol{A}_{\mathsf{q}}\,\dot{\boldsymbol{\varphi}}_l(t) + \boldsymbol{B}_{\mathsf{q}}\,\boldsymbol{\varphi}_l(t) = \boldsymbol{0}, \tag{7}$$

where $l = 1, 2$ and the real matrices $(\boldsymbol{A}_{\mathsf{q}}, \boldsymbol{B}_{\mathsf{q}})$ are given by $\boldsymbol{A}_{\mathsf{q}}{=}-\boldsymbol{H}_2 - \boldsymbol{H}_1\boldsymbol{H}_2\boldsymbol{H}_1^{-1}$ and $\boldsymbol{B}_{\mathsf{q}}{=}\boldsymbol{H}_1^2{+}\boldsymbol{H}_1\boldsymbol{H}_2\boldsymbol{H}_1^{-1}\boldsymbol{H}_2$, the sub index $\mathsf{q}$ refers to the fact that such matrices $(\boldsymbol{A}_{\mathsf{q}}, \boldsymbol{B}_{\mathsf{q}})$ encode the hamiltonian of the quantum system [8].

Dealing with a self-adjoint Hamiltonian operator, such as in closed quantum systems, it also follows that the matrix $\boldsymbol{H}$ is normal, i.e.,

$[\boldsymbol{H}, \boldsymbol{H}^\dagger] = 0$. Consequently, $\boldsymbol{H}_1$ and $\boldsymbol{H}_2$ commute, implying that matrices $\boldsymbol{A}_{\mathsf{q}}$ and $\boldsymbol{B}_{\mathsf{q}}$ take the form

$$\begin{aligned}\boldsymbol{A}_{\mathsf{q}} &= -2\boldsymbol{H}_2\\ \boldsymbol{B}_{\mathsf{q}} &= \boldsymbol{H}_1^2 + \boldsymbol{H}_2^2.\end{aligned} \tag{8}$$

Note that the fact that the Hamiltonian is normal implies, for the quantum system, that there is an orthonormal basis that makes it diagonal, while for the classical system, since $[\boldsymbol{A}_{\mathsf{q}}, \boldsymbol{B}_{\mathsf{q}}] = 0$ it implies that there are normal modes [13].

Given an initial condition $\boldsymbol{\psi}(0)$ the equation (1) has a unique solution [11], which also satisfies the second order differential equation (7) and requires another initial condition $\dot{\boldsymbol{\psi}}(0)$ that comes from $-i\boldsymbol{H}\boldsymbol{\psi}(0)$. On the other hand, both the real and imaginary parts of $\boldsymbol{\psi}$ satisfy the same equation (7). Therefore, neither of them can be neglected, as solving (7) requires knowledge of the initial conditions for $\mathrm{Re}(\boldsymbol{\psi})$ and $\mathrm{Im}(\boldsymbol{\psi})$.

Now let's turn our attention to a classical system. Considering a second-order linear system of differential equations that closely resembles to (7)

$$\ddot{\boldsymbol{q}}(t) + \boldsymbol{A}\,\dot{\boldsymbol{q}}(t) + \boldsymbol{B}\,\boldsymbol{q}(t) = \boldsymbol{0} \tag{9}$$

with $(q_1, \cdots, q_k, \cdots, q_n) = \boldsymbol{q} \in \mathbb{R}^n$ are the generalized coordinates, and $\boldsymbol{A}, \boldsymbol{B}{\in}\mathcal{M}_{n\times n}(\mathbb{R})$, i.e. the vector space of $n{\times}n$ real matrices. According to [7, 8], the equation (9) can be realized using a classical electric network. Our focus lies specifically on lumped element model circuits, where the voltage and current solely depend on time.

## IV.  SYNTHESIS OF CLASSICAL NETWORKS EQUIVALENTS TO QUANTUM SYSTEMS

An electrical network can be defined as a composite structure represented by an oriented graph, wherein each arc exhibits two time-dependent functions: current and voltage. These functions are interrelated through Kirchhoff's laws and the inherent connections that arise from the graph's depiction and the interconnections among electrical elements, such as resistors, inductors, capacitors, and more [14].

When employing methods such as node analysis, loop analysis, or pair analysis [14, 15] to determine the dynamics of a network, systems of second-order linear differential equations or integro-differential equations are obtained. However, comprehensive knowledge of the entire electrical network, including its constituent elements, their arrangement, and interconnections, is necessary for the application of these methods. As this specific information is lacking, the solution must be formulated for a generalized circuit, disregarding its graph structure or the elements present in each individual arc. Consequently, it becomes impractical to apply any of the network analysis techniques until a specific network and its corresponding arc elements are identified. Resolving this apparent circular problem calls for not the *analysis* of a single circuit but the *synthesis* of all circuits within a designated preferred family. Such prerequisites can be established based on other assumptions, which will be explored in the following sections.

The objective is to establish and characterize generalized coordinates within the electrical network, ensuring the fulfillment of equations (9). As these equations entail $n$ degrees of freedom, $n$ distinct local regions within the network where voltages or currents can be measured are defined. These regions, known as *ports*, consist of pairs of terminals that facilitate energy exchange with the surroundings and possess designated port voltages and port currents. Consequently, it can be asserted that the electrical network exhibits precisely identified, independent $n-$ports.

The proposed structure corresponds to an electrical network consisting of $n$ dipoles denoted as $\{\mathcal{N}_k\}_{k\in\mathcal{I}_n}$, also known as one$-$port networks, interconnected through an $n-$port interaction network $\mathcal{N}$ [14, 17]. Each one$-$port network introduces a port voltage or port current, corresponding to a generalized coordinate $q_k$ from $\boldsymbol{q}$ in (9), as shown in Figure 1. Since $\boldsymbol{H}$ is time-independent, the matrices $\boldsymbol{A}_{\mathsf{q}}$ and $\boldsymbol{B}_{\mathsf{q}}$ inherit this property. Consequently, the dynamics of the power grid must remain invariant under time translations, implying the absence of internal generators. The initial conditions are specified solely for each dipole, ensuring that the energy initially stored in the interaction network $\mathcal{N}$ is zero. Thus, the theory of multi-port networks can be effectively employed to synthesize $\mathcal{N}$, independent of its graph representation or constituent elements [14, 15].

Within this context, it is possible to define a transfer matrix function of $\mathcal{N}$ by taking the ratio of the Laplace transform of specific output signals to the Laplace transform of certain input signals. Depending on the chosen input and output signals, there exist four general representations: transmission, impedance, admittance, or hybrid. However, the hybrid representation for $\mathcal{N}$ must be excluded, as it would result in the mixing of input and output signals (voltages and currents) from different ports. This would compromise

the fundamental property where each network $\{\mathcal{N}_k\}_{k\in\mathcal{I}_n}$ inherently represents one and only one coordinate of the quantum system in a given basis. Consequently, the focus is on the transmission, impedance, or admittance representation of the $n-$port network $\mathcal{N}$, which is also classified as a *passive* network, meaning that the energy provided by an external source is non-negative. As previously mentioned, the initial energy is supplied by the set of one-port networks $\{\mathcal{N}_k\}_{k\in\mathcal{I}_n}$.

The dynamics of an electric network are determined by the careful application of the Kirchhoff rules, which account for the network's topology. This topology is illustrated in Figure 1, providing a schematic representation of the network.

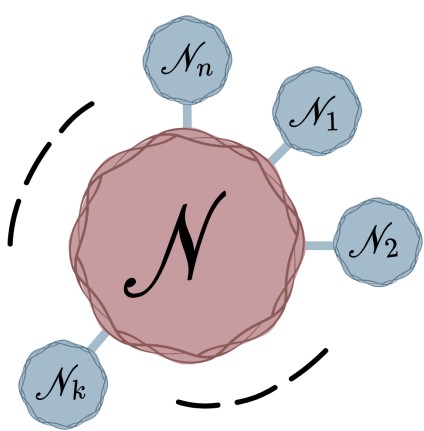

Figure 1. There are $n$ dipole networks, denoted by $\{\mathcal{N}_1,\cdots,\mathcal{N}_k,\cdots,\mathcal{N}_n\}$ interconnected through an *interaction* network $\mathcal{N}$.

Previously, the generalized coordinates in the electrical network correspond to the port voltages or port currents of each of the $n$ networks in the list $\{\mathcal{N}_k\}_{k\in\mathcal{I}_n}$. Hence, the $n-$port network $\mathcal{N}$ serves as an interaction medium, physically connecting the $n$ dipole networks. In the absence of interaction, which corresponds to disconnecting the network $\mathcal{N}$, the non-interacting case is obtained.

In quantum mechanics, a diagonal Hamiltonian $\boldsymbol{H}=diag(\lambda_1,\cdots,\lambda_n)$, excluding the trivial case where $\boldsymbol{H}=\boldsymbol{0}$, where each coordinate in Equation (1) is given by $\psi_k=\alpha_k e^{-i\lambda_k t}$, where $\alpha_k\in\mathbb{C}$. Since $\boldsymbol{H}$ is hermitian, this solution represents pure harmonic oscillations, with $\lambda_k$ being the unique natural frequency associated with the coordinate $\psi_k$. In the classical circuit, this implies that each one-port network must be non-dissipative. The reactance theorem, initially developed by Foster and subsequently by Brune [18, 19], and generalized by Cauer [20], provides a synthesis method for lossless networks. It can be concluded that the transfer function of such networks, in the complex Laplace variable $s$, has a single pole in the complex plane located on the imaginary axis and its complex conjugate. This is due to the transfer function being a *positive-real function* [18–20]. Consequently, two types of circuits can be identified for each dipole, as depicted in Figure 2.

The chosen dynamical quantity is chosen as the common variable to all these elements, $L_k$ and $C_k$, are voltages for the parallel case and current for the series case, represented in up and down part of Figure 2 respectively.

The pair of nodes $(a_k, b_k)$ must be available to connect $\mathcal{N}$, in the interaction case, however are useful, at this time, to remember that $V_k$ is the potential difference between $a_k$ and $b_k$ ($I_k$ is the current through $a_k$ and $b_k$) at up (down) case of Figure 2. In Figure 2, the voltage source $V_k(0)$ (or current source $I_k(0)$) not only represents the initial condition but also signifies that the initial energy is stored in the reactive elements. From an electromagnetic perspective, the initial potential difference across the capacitor $C_k$ corresponds to the stored electrical energy, given by $\frac{1}{2}C_k V_k^2(0)$. Similarly, the initial current in

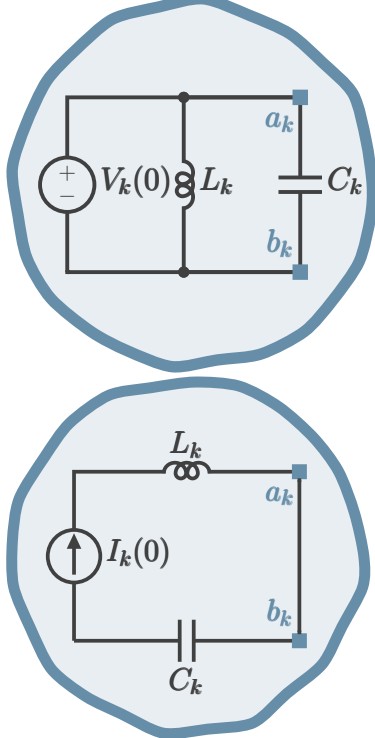

Figure 2. Alternatives topologies for each dipole sub-network $\mathcal{N}_k$. Using one of them, the signal: voltage $V_k$ (up) or current $I_k$ (down) and its initial excitation, was useful as a classical coordinate $q_k$ of (9). The above network is the dual of the one below, and vice versa.

the inductor $L_k$ represents the stored magnetic energy, given by $\frac{1}{2}L_k I_k^2(0)$. Therefore, both reactive elements serve as the initial source and contribute to the initial condition in each respective case.

An alternative way to see this conclusion directly and independently of the synthesis method employed corresponds to take the decomplexification of (1). The non$-$interacting case corresponds to $\boldsymbol{H}$ is diagonal, and from the hermiticity of $\boldsymbol{H}$ this is also real. Then the matrices $\boldsymbol{A}_{\mathsf{q}}$ and $\boldsymbol{B}_{\mathsf{q}}$ from (8) take the particular form $\boldsymbol{A}_{\mathsf{q}} = \boldsymbol{0}$ and $\boldsymbol{B}_{\mathsf{q}} = diag(\lambda_1^2, \cdots, \lambda_n^2)$, and the equation (7) is reduced to $\ddot{\boldsymbol{\varphi}}_l + \boldsymbol{B}\boldsymbol{\varphi}_l = \boldsymbol{0}$, which corresponds to a classical system whose behavior must respond to harmonic oscillator without damping. In order to compare di-

rectly with the result from the network synthesis method, let's apply the Laplace transform, $\mathcal{L}$, of the above linear differential equation and taken the $k$ component of $\boldsymbol{\varphi}_l(t)$, denoted by $x(t)$, then $X(s) = [s\,x(0) + \dot{x}(0)]/(s^2 + \lambda_k^2)$, where $X(s) = \mathcal{L}\big(x(t)\big)_{(s)}$ has only two complex conjugate poles on the imaginary axis, $\pm i\lambda_k$. The values of the inductance and capacitance of each network $\mathcal{N}_k$ can be fixed in order to satisfy $\lambda_k^2 = (L_k C_k)^{-1}$.

In summary, interaction-free corresponds to oscillation-free behavior without energy dissipation. The procedure continues to the injection of the signal $V_k$ (or $I_k$) as port-voltage (or port-current) of the interaction network. In this way the subnetworks $\{\mathcal{N}_k\}_{k \in \mathcal{I}_n}$ are ready to be connected to each port of $\mathcal{N}$. Alternatives (dual) topologies for each dipole subnetwork $\mathcal{N}_k$ for the interaction case. The pair terminal $(a_k, b_k)$ preparation of each dipole requieres that the line between $a_k$ and $b_k$ in the second (down) case of Figure 2 must be opened. For each pair $(a_k, b_k)$ must be connected to the $k-$port of $\mathcal{N}$, in order to reproduce the interaction.

To fix ideas, the port voltages of each network in the list $\{\mathcal{N}_k\}_{k \in \mathcal{I}_n}$ are chosen as generalized coordinates, $\{V_k\}_{k \in \mathcal{I}_n}$, as illustrated in the upper Figure 2. The analysis provided in [7] of the time evolution of electric circuits can be generalized here and results in a system of linear differential equations as (9).

From Kirchhoff's first law at the $\boldsymbol{k}-$node of Figure 3 one has $I_k(t) + I_{L_k}(t) + I_{c_k}(t) = 0$ where $L_k$ and $C_k$ are the mentioned tandem of the network $\mathcal{N}_k$. Given that the current-voltage relationship for the inductor $L_k$ is $V_k(t) = L_k \dot{I}_{L_k}(t)$, then $I_{L_k}(t) = I_{L_k}(\epsilon) + \frac{1}{L_k}\int_{\epsilon}^{t} V_k(u)\,du$, for $\epsilon < 0$ and $|\epsilon|$ small, with the causality condition for the

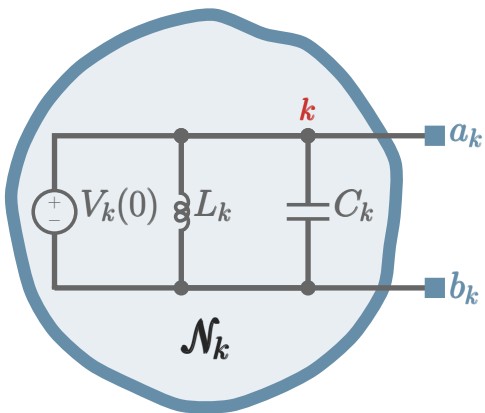

Figure 3. Under the hypothesis that the $n-$port network $\mathcal{N}$ has an *admittance representation* with admittance matrix $\boldsymbol{Y}(s)$, for each $k \in \mathcal{J}_n$: $L_k \| C_k$ tandem dipole circuit of the Figure 1 is connected to its $k-$port.

inductor current $I_{L_k}(t) = 0$, $\forall t < 0$, thus

$$I_k(t) + C_k \dot{v}_k(t) + \frac{1}{L_k}\int_\epsilon^t V_k(u)\,du = 0. \quad (10)$$

Performing a Laplace transform on (10) and using the causality condition for $k-$port voltage $V_k$, thus

$$I_k(s) + C_k s\, V_k(s) - C_k V_k(0) + \frac{1}{L_k s}\, V_k(s) = 0. \quad (11)$$

Note that the causality condition on $V_k$ is expressed here in $\mathcal{L}[\int_\epsilon^t V_k(u)du] = s^{-1}\,V_k(s)$, because $V_k(t) = 0$, $\forall t < 0$. It will be useful to define $\boldsymbol{V} = (V_1, \cdots, V_k, \cdots V_n)$ and $\boldsymbol{I} = (I_1, \cdots, I_k, \cdots I_n)$ as the voltage and current port vectors. Thus, the equation (11) can be expressed in vector form as

$$s\boldsymbol{\mathcal{V}}(s) - \boldsymbol{V}(0) + s^{-1}\boldsymbol{\omega}_0^2\boldsymbol{\mathcal{V}}(s) + \boldsymbol{C}^{-1}\boldsymbol{Y}(s)\boldsymbol{\mathcal{V}}(s) = \boldsymbol{0} \quad (12)$$

where $\boldsymbol{\mathcal{V}}(s) = \mathcal{L}[\boldsymbol{V}(t)]$ and $\boldsymbol{C}$ and $\boldsymbol{L}$ are diagonal matrices which contain the capacitors and inductors of the one$-$port networks of the list $\{\mathcal{N}_k\}_{k=1,\cdots,n}$, $\boldsymbol{C} = diag(C_1, \cdots, C_n)$ and $\boldsymbol{L} = diag(L_1, \cdots, L_n)$. In this way, $\boldsymbol{\omega}_0^2 = (\boldsymbol{L}\boldsymbol{C})^{-1}$ contains the proper frequencies of each $L_k \| C_k$ port tandem of the Figure 2. Additionally, the

transfer relation coming from the interaction network $\mathcal{N}$ is utilized: $\boldsymbol{\mathcal{I}}(s) = \boldsymbol{Y}(s)\boldsymbol{\mathcal{V}}(s)$. Applying the inverse Laplace transform ($\mathcal{L}^{-1}$) on (12) to obtain the equation in the time domain

$$\dot{\boldsymbol{V}}(t) + \boldsymbol{\omega}_0^2\int_0^t \boldsymbol{V}(u)du + \boldsymbol{C}^{-1}\mathcal{L}^{-1}[\boldsymbol{Y}(s)\boldsymbol{\mathcal{V}}(s)]_{(t)} = \boldsymbol{0}$$

and differentiate with respect to $t$ to obtain a second-order differential equation

$$\ddot{\boldsymbol{V}}(t) + \boldsymbol{\omega}_0^2\boldsymbol{V}(t) + \boldsymbol{C}^{-1}d_t\left\{\mathcal{L}^{-1}[\boldsymbol{Y}(s)\boldsymbol{\mathcal{V}}(s)]_{(t)}\right\} = \boldsymbol{0}. \quad (13)$$

where $d_t$ is a compact notation of the usual time derivative. The matrix elements of $\boldsymbol{Y}(s)$ are rational functions: quotients of polynomials in $s$. A necessary and sufficient condition for that the equation (13) has the form of (9) is

$$\boldsymbol{Y}(s) = \boldsymbol{\alpha} + \frac{1}{s}\boldsymbol{\beta} \quad (14)$$

where the constant matrix $\boldsymbol{\alpha}$ and $\boldsymbol{\beta}$ can be synthesized using the general method exposed in [16, 17]. Finally, the matrices $(\boldsymbol{A}, \boldsymbol{B})$ of (9) are written in terms of admittance matrices $(\boldsymbol{\alpha}, \boldsymbol{\beta})$ of the interaction network $\mathcal{N}$ and $(\boldsymbol{L}, \boldsymbol{C})$ from $k-$tandem $L_k \| C_k$ circuit, which constitutes the subnetwork $\mathcal{N}_k$ showed in Figure 3, as

$$\boldsymbol{A} = \boldsymbol{C}^{-1}\boldsymbol{\alpha},$$
$$\boldsymbol{B} = \boldsymbol{C}^{-1}\boldsymbol{\beta} + \boldsymbol{\omega}_0^2. \quad (15)$$

Using the *synthesis* methods [14–17] on the general topology proposed in Figure 1, it is possible to obtain an interaction network $\mathcal{N}$ whose admittance matrices define two matrices from (15) corresponds to the two matrices from (8)

$$(\boldsymbol{A}, \boldsymbol{B}) \asymp (\boldsymbol{A}_{\mathfrak{q}}, \boldsymbol{B}_{\mathfrak{q}}). \quad (16)$$

In other words these classical matrices can be mapped to the quantum hamiltonian. In other

words, $\boldsymbol{H}$ is codified in terms of $\boldsymbol{A}$ and $\boldsymbol{B}$, in particular in $\boldsymbol{\alpha}$ and $\boldsymbol{\beta}$ of the synthesis of the interaction network $\mathcal{N}$. In general, the proposed configuration in the Figure 1 each port voltage $V_k$ works as the real, or imaginary, part of the $k-$component of the wave function $\boldsymbol{\psi}$.

A similar procedure can be repeated using the *impedance representation* of the network $\mathcal{N}$, simply by interchanging the following quantities: voltages by currents, inductances by capacitances, conductances by resistances in order to obtain identical equations to (9) so that now the generalized coordinates are the port$-$currents $\boldsymbol{I}$. Note that this alternative not only gives us a plan B in the implementation but also both implementations could be done simultaneously, so that the port$-$voltages of one and the port$-$currents of the other are represents exactly, i.e. is associated through $\asymp$, the real and imaginary part of the wave function $\text{Re}(\boldsymbol{\psi}) \asymp \boldsymbol{V}$ and $\text{Im}(\boldsymbol{\psi}) \asymp \boldsymbol{I}$ [8].

An alternative and more formal approach is provided by the analytic representation of a given signal, e.g. a potential. This representation extends the concept of *phasor* associated to signals of a fixed frequency, in the sense that it allows representing signals whose amplitude and phase are time variable functions [24]. For a given real signal $x(t)$, its *analytic signal* is another time variable function $x_\alpha(t)$, defined by

$$x_\alpha := x + iy, \tag{17}$$

where $y := \mathcal{H}[x]$ is the Hilbert transform of $x$. Then $x = \text{Re}(x_\alpha)$ and $y = \text{Im}(x_\alpha)$. The reference to $x$ in $x_\alpha$ must be maintained because the analytic signal is a complex representation of a real signal $x$. Conversely, for an analytic function $f(z)$ in the upper half-plane of $\mathbb{C}$, and a real function $x$ such that $x(t){=}\text{Re}\big(f(t{+}0{\cdot}i)\big)$,

then $\text{Im}\big(f(t{+}0{\cdot}i)\big){=}\mathcal{H}[x(t)]$ up to an additive constant, provided this Hilbert transform exists. For more details, Titchmarsh [21] conducted a rigorous mathematical analysis of Hilbert transforms in relation to analytic functions, Guillemin [22] later derived equivalent formulas for Hilbert transforms, followed by Oswald [23] who established a connection between analytic signals and Hilbert transforms. A modern treatment of such ideas can be found in [24, 25].

Given a classical network such that the potential *real* signal $\boldsymbol{V}$ is associated with the real part of the wave function $\boldsymbol{\psi}$, its imaginary part can be obtained by calculating the Hilbert transform of $\boldsymbol{V}$, thus

$$\begin{aligned}\text{Re}(\boldsymbol{\psi}) &\asymp \boldsymbol{V}, \\ \text{Im}(\boldsymbol{\psi}) &\asymp \mathcal{H}[\boldsymbol{V}].\end{aligned} \tag{18}$$

Each component $\psi_k$, which represents the probability amplitudes, is associated with a classical quantity, e.g. $V_k$, in this way

$$\boldsymbol{\psi} \asymp \boldsymbol{V}_\alpha {=} \boldsymbol{V} {+} i\,\mathcal{H}[\boldsymbol{V}]. \tag{19}$$

There is a connection between the concept of envelopes of real waveforms with respect to it Hilbert transform [33], that allow us to express the envelope of $x$, namely env$[x]$, as

$$\texttt{env}[x] = \sqrt{x^2 + \big(\mathcal{H}[x]\big)^2}. \tag{20}$$

Regarding the described quantum system on numerable Hilbert space, the modulus of each component $\psi_k$ squared represents the probability, $|\langle\beta_k|\psi\rangle|^2$, of the system's state $|\psi\rangle$ being $|\beta_k\rangle$, according to the Born rule. In this way, this probability, $p_k$, corresponds to the envelope of the classical signal squared

$$p_k \asymp \big(\texttt{env}[V_k]\big)^2, \tag{21}$$

note that $p_k{=}|\langle\beta_k|\psi\rangle|^2$ is a *probability mass function* for the state level variable $k$. A concrete construction of such electrical network

with their corresponding experimental measurements can be found in [8].

## V. ELECTRICAL PAULI REPRESENTATION

Pauli matrices [26] are one of the most significant and widely recognized sets of matrices in the realm of quantum physics. They are particularly crucial in both physics and chemistry, being used for quantum simulations [29, 30] and to describe Hamiltonians of many-body spin glasses [27, 28]. From the mathematical point of view, Pauli matrices $\{\boldsymbol{\sigma}_k\}_{k=1,2,3}$, together with the identity matrix $\boldsymbol{\sigma}_0$, form a basis for the vector space $\mathcal{M}_{2\times 2}(\mathbb{C})$, which is fundamental in quantum theory, as they are used to represent the observables of a two-level quantum system. However, for systems of higher dimension, it is possible to construct the appropriate Hamiltonian using the tensor product of such matrices [31].

Let's see now how to find the explicit classical circuit corresponding to the Hamiltonian $\boldsymbol{H}=\sum_{k=0}^{3}\xi_k\boldsymbol{\sigma}_k$, note that $\boldsymbol{H}=\boldsymbol{H}^{\dagger}$ thus $\xi_k\in\mathbb{R}$. As a gift, a classical circuit corresponding to each Pauli operator will be obtained by considering $(\xi_1,\xi_2,\xi_3)$ equal to any element of $\{(1,0,0),(0,1,0),(0,0,1)\}$. Explicitly

$$\boldsymbol{H} = \begin{pmatrix} \xi_0 + \xi_3 & \xi_1 - i\,\xi_2 \\ \xi_1 + i\,\xi_2 & \xi_0 - \xi_3 \end{pmatrix} \qquad (22)$$

From (8)

$$\boldsymbol{A}_{\mathfrak{q}} = \begin{pmatrix} 0 & 2\xi_2 \\ -2\xi_2 & 0 \end{pmatrix},$$

$$\boldsymbol{B}_{\mathfrak{q}} = \begin{pmatrix} (\xi_0+\xi_3)^2+\xi_1^2-\xi_2^2 & 2\xi_0\xi_1 \\ 2\xi_0\xi_1 & (\xi_0-\xi_3)^2 + \xi_1^2-\xi_2^2 \end{pmatrix}.$$

In section IV, it is stated that in the non-interacting case of a quantum system, when the Hamiltonian $\boldsymbol{H}$ is diagonal, it corresponds to disconnect the interaction network $\mathcal{N}$, i.e. $\boldsymbol{Y} = \boldsymbol{0}$. From (15) obtain $\boldsymbol{A}=\boldsymbol{0}$ and $\boldsymbol{B}=\boldsymbol{\omega}_0^2$. Given that $\boldsymbol{A}\asymp\boldsymbol{A}_{\mathfrak{q}}$ then $\xi_2 = 0$ and from $\boldsymbol{B}\asymp\boldsymbol{B}_{\mathfrak{q}}$ then $\xi_1 = 0$. Note that $\xi_0 \neq 0$ in order to include the possibility of different natural frequencies. In this way the first association is $(L_1C_1)^{-1}\asymp(\xi_0+\xi_3)^2$ and $(L_2C_2)^{-1}\asymp(\xi_0-\xi_3)^2$.

Now, connecting the interaction network $\boldsymbol{Y}(s) = \boldsymbol{\alpha}+(1/s)\boldsymbol{\beta}$, a two-gate network is synthesized to a gyrator [32] connected in parallel with a network of inductors with a $\prod$ structure [16, 17]. The matrices of $\boldsymbol{Y}$ are given by

$$\boldsymbol{\alpha}=\begin{pmatrix} 0 & g \\ -g & 0 \end{pmatrix}, \ \boldsymbol{\beta}=\begin{pmatrix} L_a^{-1} + L_b^{-1} & -L_c^{-1} \\ -L_c^{-1} & L_a^{-1} + L_b^{-1} \end{pmatrix}.$$

where $g$ is the conductance parameter of the gyrator.

Figure 4 shows the complete circuit associated with the two-level quantum system whose Hamiltonian is given by (22).

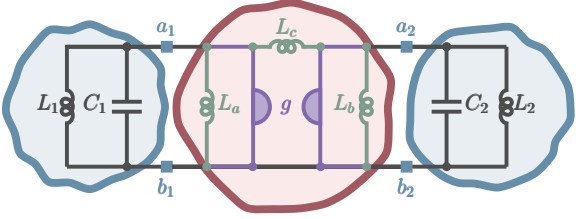

Figure 4. The *admittance* representation of the complete circuit whose port−voltages $(V_1, V_2)$ reproduce the time evolution (1) of the quantum system given by the hamiltonian (22). In order to simplify the figure, the initial condition $V_1(0)$ and $V_2(0)$ has been suppressed.

The electric Pauli representation comes from (16), the first observation is that $C_1=C_2$, hereinafter denoted by $C$. From

$(\boldsymbol{A}, \boldsymbol{B}) \asymp (\boldsymbol{A}_\mathsf{q}, \boldsymbol{B}_\mathsf{q})$ obtain

$$(CL_1)^{-1} \asymp (\xi_0 + \xi_3)^2$$
$$(CL_2)^{-1} \asymp (\xi_0 - \xi_3)^2$$
$$(CL_a)^{-1} + (CL_b)^{-1} \asymp \xi_1^2 - \xi_2^2 \qquad (23)$$
$$-(CL_c)^{-1} \asymp 2\xi_0\xi_1$$
$$C^{-1}g \asymp 2\xi_2$$

Under an equivalent circuital point of view, the pair of inductors $(L_1, L_a)$ and $(L_2, L_b)$ are arranged in parallel. In order to build the equivalent circuit, we can unify the pairs of inductors arranged in parallel: $L_1^* = L_1 \| L_a$ and $L_2^* = L_2 \| L_b$. On the quantum side, the number of independent parameters are exactly four $(\xi_0, \xi_1, \xi_2, \xi_3)$, finally on the classic side, the number of independent parameters are also four $(L_1^*, L_2^*, L_c, g)$, because all the parameters can be rescaled as they are all affected by the same $C$.

## VI. CONCLUSION AND FINAL OBSERVATIONS

The aim of present work it was to prove that there is a way to simulate the time evolution of a given quantum system on finite-dimension Hilbert space, using electrical networks.

Regarding the simulation of a quantum system $\mathbb{Q}'$, the objective is to find another system that closely imitates the behavior of $\mathbb{Q}$ as accurately as possible. In other words, a *casting call* of quantum systems or *actors* must be conducted, which can be quite limited due to the challenging task of finding a suitable candidate to simulate $\mathbb{Q}'$. It is worth noting that the mapping $\boldsymbol{\Omega}_{\boldsymbol{\omega}}(\boldsymbol{H}) := \boldsymbol{\omega}\boldsymbol{H}\boldsymbol{\omega}^{-1} + i\dot{\boldsymbol{\omega}}.\boldsymbol{\omega}^{-1}$ is a generalization of a similarity transformation [3]. By leveraging the fact that all these quantum systems are equivalent through the

mapping $\boldsymbol{\Omega}_{\boldsymbol{\omega}}$, which connects a given Hamiltonian $\boldsymbol{H}$ to any other Hamiltonian $\boldsymbol{H}'$ [2], a well-defined path emerges for the implementation of quantum system simulation using classical circuits.

Given a quantum system $\mathbb{Q}$ over a Hilbert space of finite dimension $n$, there is a general topology, as shown in Figure 1, of $n-$subnetworks $\{\mathcal{N}_k\}_{k=1,\cdots,n}$ connected to an interaction network $\mathcal{N}$ of $n-$ports. Such a network $\mathcal{N}$ can be adequately synthesized in such a way that the voltages (or currents) of each of its ports evolves in time in the same way as the real (imaginary) part of the original quantum system. In other words, the time evolution of such a quantum system can be analogically simulated from a classical tuned circuit. To simulate a two-state quantum system, e.g. a qubit, a two-port network will be needed. To simulate an $n-$state quantum system, a network of $n-$ports will be needed.

Electric circuits under alternating current have recently demonstrated the ability to simulate various topological phenomena in physics. This type of classical constructions allows us to deal with infinite degrees of freedom encoded along a transmission line. Following these ideas, it remains for future work to formalize them in terms of [2] to identify precisely how to map the dynamics of spatially distributed electrical networks to quantum systems of infinite degrees of freedom. In order to continue with this classical approach, one could choose to keep the electrical branch working with transmission lines [34, 35] or switch to an optical approach [36].

Recall that a quantum computer, e.g. Google or IBM, that possess $N$ qubits can encode $2^N$ states in order to process information, provided that: the number of logical qubits is $N$ and the decoherence times are greater than

the execution time of the algorithm. While there are many challenges to implementing the classical ideas discussed in this article, one of the advantages of these is that the above conditions are not required.

Regarding Born's rule, it has been possible to reinterpret it based on the envelope of the port-potentials of each subnetwork as (21).

A final comment on this matter could be the advantage of employing these classical systems is their present controllability, which allows for precise manipulation of their temporal evolution. By incorporating classical actors into the catalog of quantum system simulators, traditionally not involved in such simulations, the scope and potential of simulating the quantum system $\mathbb{Q}$ is broadened. This work significantly contributes to the advancement of quantum simulations.

## ACKNOWLEDGMENTS

This work was supported by the European Union, Next Generation UE/MICIU/Plan de Recuperacion, Transformacion y Resiliencia/Junta de Castilla y Leon. We thank to `FIDESOL`, `UGR` and `UNIR` for the support and recall also the anonymous readers for their constructive criticism to this work.

## COMPETING INTEREST

The author declares that there are no competing interests.

## DATA AVAILABILITY STATEMENT

There is no data associated in the manuscript.

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
