# Peer review of "Classical circuits can simulate quantum aspects"

_SciPost Physics_

## Round 1 · Referee Report · Anonymous · 2024-9-8

Strengths

• The paper presents a robust theoretical foundation by establishing a generalized similarity transformation that links different Hamiltonians. This allows classical circuits to simulate the time evolution of quantum systems, including the Schrödinger equation, providing a new approach to quantum simulations.
• The work introduces an innovative method for modelling quantum systems using classical electrical circuits. By constructing equivalent classical networks, the paper demonstrates how the dynamics of quantum systems can be replicated with classical components considering the voltages and currents.
• The methodology supports scalability from 2-state to n-state quantum systems. This versatility enables the simulation of increasingly complex quantum phenomena using classical circuits.
• The author provides a theoretical recipe for constructing classical electrical circuits capable of solving the Schrödinger equation for various quantum systems.
• A key innovation is the reinterpretation of Born’s rule in the context of electrical circuit simulations. This novel perspective connects classical signals to quantum probabilities, broadening the scope of quantum simulations using classical networks.

Weaknesses

• The paper suffers from problems in writing style and grammar. Some sections are poorly structured, making it difficult for readers to follow the logical flow of the arguments and the technical description.

• While the theoretical framework is solid, but the absence of detailed empirical results makes it hard to assess the practical feasibility of using classical circuits for quantum system simulations. Perhaps some references to simulations could be included.

• The paper could benefit from a deeper discussion of the limitations or potential challenges in implementing the proposed methods, especially regarding scalability and the precision of classical simulations for more complex quantum systems.

Report

The manuscript presents an interesting approach to simulating quantum systems using classical electrical circuits. The author introduces a generalized similarity transformation that links different Hamiltonians, enabling quantum systems to be simulated through classical networks. This work presents significant results for theoretical physics and practical applications, offering an accessible method for quantum simulations without requiring quantum computers. Given the topic and the scope of Scipost Physics, I recommend the paper for publication, pending minor revisions.

Requested changes

- Review the text carefully for typos and grammar errors.

- Page 1, first paragraph to the end. Please define intersimulatable better, as it seems to be a definition proposed by the author, which requires a clearer definition.

- Could you replace "realized" in the last paragraph on page 2 with "simulated"?

- Page 5, caption of the figure. Please include the Figure Title “Electrical network” at the beginning of the caption.

- Please, place the word Equation or Eq. before each reference to equations in the text in a uniform manner, as it is messy sometimes. It has it and sometimes it doesn't. It isn't very clear to follow the text.

- On page 6, please rewrite the last sentence of the paragraph that begins with: “In summary”, as it is written incomprehensible.

- On page 8, the second paragraph that begins with “A similar…” Please rewrite from “Note that…” That paragraph is also unclear.

- Figure 4: Clarify in the caption what the circles made on the circuit represent.

Recommendation

Publish (easily meets expectations and criteria for this Journal; among top 50%)

---

## Editorial Decision

in_refereeing